# Microbial Diversity and Metabolites Dynamic of Light-Flavor *Baijiu* with Stacking Process

**Zhaojie Li** [1,2]**, Yi Fan** [1,2]**, Xiaoning Huang** [1,2] **and Beizhong Han** [1,2,*]

1   College of Food Science and Nutritional Engineering, China Agricultural University, Beijing 100083, China; lzjcau@126.com (Z.L.); fanyi628@126.com (Y.F.); hxning926@sina.com (X.H.)
2   Key Laboratory of Viticulture and Enology, Ministry of Agriculture and Rural Affairs, Beijing 100083, China
*   Correspondence: hbz@cau.edu.cn

**Abstract:** Stacking is a widely used method of microbial enrichment in the field of fermentation and is traditionally used to promote flavor in Chinese sauce-flavor *Baijiu*; however, its precise mechanism is unknown. This study assessed the fermentation process of light-flavor *Baijiu* with the simplest microbial source. After comparing differences in the microbial composition of different kinds of *Daqu*, a high-temperature *Daqu* with a microbial composition that significantly differs from light-flavor *Daqu* was selected for stacking. The physical and chemical indicators, microbial community composition, and metabolite profiles during the fermentation process were tracked, and the total ester content in *Baiju* was significantly higher with stacking than *Baijiu* without stacking. The dominant bacteria during stacking fermentation were *Bacillus* and *Enterococcus*, while *Lactobacillus* was the dominant bacteria during middle and late fermentation periods. Low levels of *Lactobacillus* and *Pichia* in *Daqu* were screened and enriched during the stacking process, while the glucose and acetate content significantly increased. Flavor compounds such as esters and acids were positively correlated with dominant genera such as *Lactobacillus*, *Bacillus*, and *Pichia*. Stacking provides microorganisms for environmental screening, which regulates the microbial community structure and produces various metabolites and precursors of flavor substances to fully saccharify and promote the production of flavor substances. Stacking during the production of light-flavor *Baijiu* can help regulate the fermentation process and improve *Baijiu* quality.

**Keywords:** stacking fermentation; *Baijiu*; flavor accumulation; microbial composition

## 1. Introduction

Stacking fermentation is also known as piling or composting and is a common method of microbial enrichment and fermentation. Stacking is widely used in composting, sewage treatment, environmental engineering, food fermentation, and other fields [1,2]. The goal of stacking fermentation is to increase the biomass of microorganisms under high-temperature and high-humidity conditions, which can enrich microorganisms. The substrate is fermented and decomposed by microorganisms to produce organic fertilizer and other products [3]. Diverse microorganisms from complex microbial communities are involved in stacking fermentation. Identifying the succession and function of complex microbial communities is key to revealing the mechanisms of stacking fermentation, which largely remain unclear.

*Baijiu* is the traditional Chinese distilled liquor with the mixed-species fermented starter called *Daqu*. The alcoholic fermentation processing of *Baijiu* is showed in Figure 1a. Sorghum as the raw materials is soaked for 18–20 h, then steamed, cooled, and mixed with *Daqu* for alcoholic fermentation. Stacking fermentation is the unique traditional process that makes sauce-flavor *Baijiu* (Chinese liquor) during the alcoholic fermentation process, and plays a significant role in increasing the biomass of microorganisms and balancing chemical substances in the fermented grains [4]. Stacking fermentation usually happens

before mixing with *Daqu* for alcoholic fermentation. Thus, the original *Daqu* in Figure 1 means the *Daqu* for alcoholic fermentation while stacking *Daqu* as the *Daqu* for stacking fermentation process. Stacking *Daqu* is mixed in after the sorghum is steamed and cooled, after which the grains are piled together and naturally fermented. The inside of the pile begins to heat first and then radiates outwards. After three to four days, temperatures can reach 60–65 °C, which is known as high-temperature stacking fermentation [5]. Stacking fermentation typically lasts for three to five days, after which the fermented grains are poured into a pit and the alcoholic fermentation process begins. Current research on stacking fermentation has outlined patterns of microbial diversity during both the stacking fermentation and alcoholic fermentation periods. During the stacking fermentation period, microbial diversity and the total population of filamentous fungi (such as *Paecilomyces variotii* and *Aspergillus oryzae*) significantly increased. This leads to an increase in amylase activity and the promotion of starch utilization [4]. *Bacillus, Lentibacillus, Thermomyces,* and *Thermoascus* are the dominant microbial genera during stacking fermentation of sauce-flavor *Baijiu,* and succession during the alcoholic fermentation process [6].

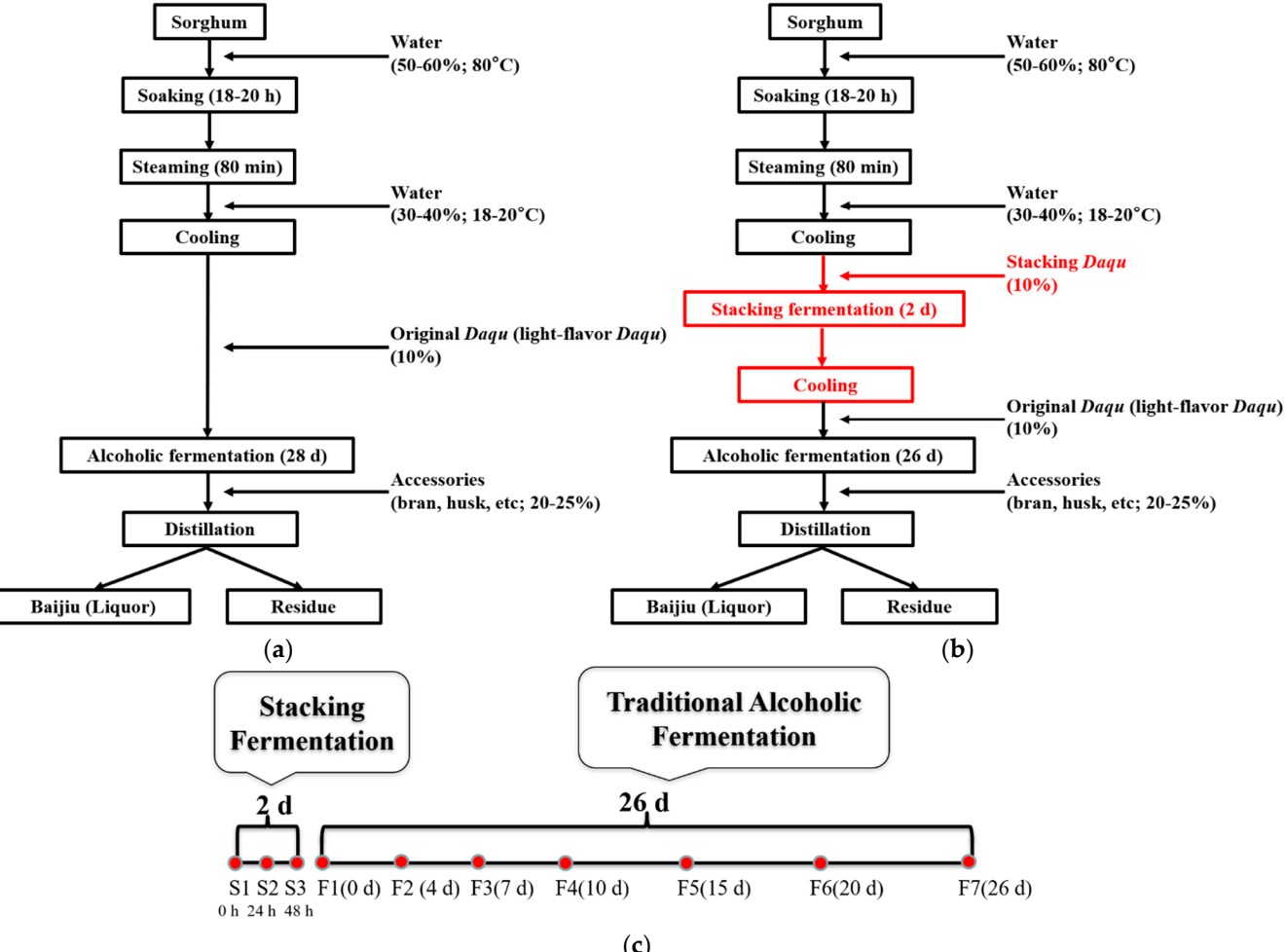

**Figure 1.** (**a**) Process of light-flavor *Baijiu*; (**b**) Fermentation process with stacking used in this study (the red part shows when the stacking fermentation applied in this study); (**c**) Sampling points of fermentation in this study.

The microbial mechanism of stacking fermentation entails the enrichment of microorganisms in high-temperature *Daqu* to produce sauce-flavor *Baijiu,* which can be considered a disturbance to the stability and function of the microbial community during the alcoholic fermentation process. Most research on stacking fermentation has focused on microbial di-

versity and succession patterns, but the mechanism of stacking fermentation is unclear [7,8]. As one of the three major *Baijiu* flavors, sauce-flavor *Baijiu* has the highest content of flavor compounds such as organic acids and esters [5]. Since stacking fermentation contributes to flavor substances, it has been incorporated into the fermentation process of other flavor types, such as sesame-flavor *Baijiu* [9,10]. It is unclear whether stacking fermentation can be used to increase the flavor content of other flavors of *Baijiu*, and the mechanism and significance of improving these flavor compounds are not yet known.

In recent years, the quality of liquor produced by traditional fermentation techniques has declined due to climatic and environmental changes. A lack of flavor substances is the most prominent problem. Researchers have also used fortified *Daqu* and inoculated ester-producing bacteria to strengthen flavor during fermentation [11–13]. Among all the flavor types of *Baijiu* fermentation process, the light-flavor *Baijiu* has the clearest microbial sources and the simplest fermentation process. It has a short fermentation cycle, and the majority of its microorganisms come from *Daqu*. Light-flavor *Baijiu* fermentation is an ideal model to explore the mechanism of stacking fermentation and its possible applications.

In this study, we researched the fermentation process of light-flavor *Baijiu* as a research model to explore the application and microbial mechanisms of stacking fermentation. We used amplicon sequencing to compare differences in the microbial composition of different *Daqu* samples and performed stacking fermentation on the *Daqu* sample that was most different from light-flavor *Daqu*. The physical and chemical indicators, microbial community composition and diversity, and flavor compounds were tracked during the stacking and alcoholic fermentation process, and the total acid and total ester content of the final distilled liquor was detected. Our study evaluates the stacking fermentation process from multiple perspectives, highlights the impacts of stacking fermentation on the alcohol fermentation process, reveals the mechanism behind stacking fermentation, and proposes several potential applications.

## 2. Materials and Methods

### 2.1. Daqu Sampling and Stacking Fermentation

Medium- and high-temperature *Daqu* were selected from four large-scale *Baijiu* production operations in the four largest *Baijiu* production provinces (*Sichuan, Henan, Shandong,* and *Jiangsu*). The *Daqu* samples were ground, after which 100 g was weighed into a sterile bag and stored at −20 °C (see Table 1 for details).

**Table 1.** Detailed information of *Daqu* samples in this study.

| *Daqu* Samples * | Raw Material | Location (Province) |
| --- | --- | --- |
| 1-LI-L | Barley, pea | *Shanxi* |
| 2-ST-MH | Wheat | *Henan* |
| 3-ST-MH | Wheat | *Jiangsu* |
| 4-SA-H | Wheat | *Sichuan* |
| 4-ST-H | Barley, wheat | *Sichuan* |
| 5-ST-M | Wheat | *Shandong* |
| 5-ST-MH | Wheat | *Shandong* |
| 5-SE-H | Wheat | *Shandong* |

* the number designates the name of liquor factory; the letters "LI", "ST", "SA", and "SE" indicate the flavor of *Daqu* (short for Light, Strong, Sauce, and Sesame (miscellaneous-flavor), respectively; the letters "H", "MH", "M", and "L" indicate the maximum fermentation temperature of the *Daqu*-making process (short for high temperature *Daqu* 60–70 °C, medium-high temperature *Daqu* 55–65 °C, medium temperature *Daqu* 50–60 °C, and low temperature *Daqu* 40–50 °C, respectively).

The test fermented grain samples were collected from three batches in the experimental workshop of the *Fenjiu* Group (*Fenyang* City, *Shanxi* Province). The fermentation process flow diagram is shown in Figure 1b. A total of 60% hot water (80 °C) was added to sorghum and soaked for 20 h. Then steamed for 80 min and added 40% water (room temperature). After cooling into room temperature, 10% stacking *Daqu* was added and mixed for 2 days stacking process. After cooling, 10% original *Daqu* (1-LI-L) was added and mixed for

26 days alcoholic fermentation. The 28 days fermentation process can be divided into two stages: 2 days stacking fermentation and 26 days traditional alcoholic fermentation. Microorganisms active during the fermentation process can rapidly increase temperatures, so three samples were taken in a short period (every 24 h): S1, S2, and S3. However, microbial changes during the traditional alcoholic fermentation process are relatively slow, and the rate of temperature rise significantly decreased. Based on previous research on the light-flavor fermentation process, seven samples were collected at 0 days, 4 days, 7 days, 10 days, 15 days, 20 days, and the end of the fermentation process: F1, F2, F3, F4, F5, F6, and F7, respectively (Figure 1b). A total of 500 g fermented grain samples were randomly obtained from each sampling point and packed into two sterile bags. One was stored at −20 °C, and the other was immediately subjected to physical and chemical index detection and microbial culture.

### 2.2. Amplicon Sequencing and Analysis

Microbial DNA was extracted from *Daqu* and fermented grain samples via genomic DNA extraction from soy sauce koji samples [14]. The reported bacterial universal primers 515F (5′-GTGCCAGCMGCCGCGGTAA-3′) and 806R (5′-GGACTACHVGGGTWTCTAAT-3′) were used to amplify the V4 variable region of the bacterial 16S rRNA gene [15,16], and the fungal universal primers ITS1F (5′-CTTGGTCATTTAGAGGAAGTAA-3′) and ITS1R (5′-GCTGCGTTCTTCATCGATGC-3′) were used to amplify the ITS region of the fungal rRNA gene [17]. Each sample was subjected to three PCRs with different annealing temperatures. Mixed products were purified by 2% agarose gel electrophoresis and recovered with the GeneJET gel recovery kit (Thermo Fisher Scientific, Wilmington, DE, USA), while the resulting products were subjected to sequencing on an Illumina HiSeq platform provided by *Beijing* Ovison Gene Health Technology Co., Ltd. The obtained representative sequences of all operational taxonomic units (OTUs) were compared with Genbank (bacterial 16S rRNA gene sequence database https://www.ncbi.nlm.nih.gov/, accessed on 2 February 2022) and Greengenes (fungal rRNA gene ITS region sequence database http://greengenes.secondgenome.com, accessed on 2 February 2022) to determine the phylum, class, order, family, and genus of all OTUs.

### 2.3. Physical and Chemical Properties and Microbial Count

Thermometers were placed in the middle of the upper, middle, and lower three layers of the stack and the pit for measurement, and data was imported into the computer for statistical analysis [18,19]. The moisture content of fermented grain samples was determined using the direct drying method as the reference [20]. A 5 g sample was weighed and dried to a constant weight in a 105 °C drying oven, after which it was cooled in a desiccator for 0.5 h before weighing. The starch, sugar, acidity, and alcohol content in fermented grain samples was determined by near-infrared spectroscopy [20].

Total count of mesophilic aerobic bacteria (TMAB) was cultured by pouring plate counting medium, after which it was incubated at 30 °C for 3 days and counted. Lactic acid bacteria (LAB) were cultured in MRS medium, cultured at 30 °C for 4 days, and counted. Before pouring the medium, 0.2% Natamycin was added. A total of 100 μL of the sample microbial suspension was coated and cultured (containing chloramphenicol) in red bengal selective medium under aseptic conditions, cultured at 30 °C for 5 days, and counted.

### 2.4. Volatile Flavor Substances and Non-Volatile Metabolites

A total of 10 μL of internal standard 2-methyl-3-pentanone (81 mg/kg methanol) was added to a 3.0 g mash sample and sealed in a 15 mL headspace sample bottle in a 40 °C water bath for 30 min. The 50/30 um DVB/CAR/PDMS extraction head was inserted into the headspace sample bottle for adsorption for 30 min (in 40 °C water), after which the extraction head was inserted into the GC inlet for 2 min for analysis. As a reference, the GC conditions were the same [21]. Total ester determination method was the same as the reference [20].

A total of 200 mg of the fermented grain sample was placed in a 2 mL centrifuge tube, after which 1.5 mL of ultrapure water (refrigerated at 4 °C) was added. It was then ground with a Mini-Beadbeater for 30 s and placed in an icebox to cool for 5 min. It was again ground for 30 s, cooled for 5 min, and centrifuged at 13,000 rpm at 4 °C for 10 min. A total of 1.0 mL of the supernatant was put in a new centrifuge tube, an equal amount of heavy water phosphate buffer was added, and it was centrifuged at 4 °C, 13,000 rpm for 10 min. The supernatant (not less than 600 µL) was placed in an NMR tube with an inner diameter of 5 mm. $^1$H NMR test parameter settings and raw data analysis methods were the same as the reference [21].

### 2.5. Statistical Analysis of Data

O2PLS was used to analyze the correlation between microorganisms and other indicators during the fermentation process. In this study, physical and chemical indicators were considered independent variable X and microorganisms as dependent variable Y. The microorganisms were considered independent variable X and metabolites as dependent variable Y to calculate the correlation coefficient ($\rho$) between any two variables in X and Y. Cytoscape (v.3.4.0) software was used to visualized and analyze $|\rho| \geq 0.7$. Principal co-ordinates analysis (PCoA) were performed and visualized using the vegan package and the gggplot2 package in R. Heatmaps of differential metabolites were constructed using the pheatmap package in R.

## 3. Results

### 3.1. The Selection of Daqu for Stacking Fermentation

To identify suitable *Daqu* samples for stacking fermentation, we analyzed the microbial diversity of different kinds of *Daqu*. In this study, we used amplicon sequencing to investigate the microbial community structure of different kinds of *Daqu* and compare them with the original light-flavor *Daqu*. A total of 533,783 bacterial 16S rRNA gene reads and 430,503 fungal ITS reads passed the quality test and were obtained. The rarefaction curves of the samples demonstrated that the sequencing depth and quality of the sequencing results are sufficient. Therefore, we were able to identify the true microbial community structure in each sample (Figure S1).

Taxonomic annotation of OTUs was generated by bacterial 16S rRNA gene clustering in the eight *Daqu* samples shown in Figure 2a. We found that 101 OTUs belonged to 39 families and 50 genera (all OTUs were identified at the family level, 93 OTUs were identified at the genus level, and 38 OTUs were identified at the species level). Of these, 20 genera had an abundance exceeding 1%, which directly reflects the microbial community structure in the samples. The dominant bacteria in *Daqu* samples belonged to the following genera: *Erwinia*, *Staphylococcus*, *Thermoactinomyces*, *Lactobacillus*, and *Lactococcus*. Of them, *Thermoactinomyces* is a high-temperature actinomyces and accounted for a high proportion of samples 3-ST-MH, 4-SA-H, and 4-ST-H. The relatively high abundance of these high-temperature-resistant microorganisms is related to the relatively high temperature during *Daqu* making. *Erwinia* and *Lactobacillus* are the dominant microorganisms in 1-LI-L and 5-ST-M, respectively, where the temperatures during *Daqu* production are relatively low. PcoA analysis, as shown in Figure 2c, demonstrated that the bacterial community structure of 4-SA-H, 4-ST-H, and 5-SE-H samples differ the most from 1-LI-L samples (light-flavor *Daqu*). We identified *Daqu* with a different community structure from 1-LI-L (light-flavor *Daqu*) for stacking fermentation to increase microbial diversity during the alcoholic fermentation process: *Daqu* 4-SA-H, 4-ST-H, and 5-SE-H.

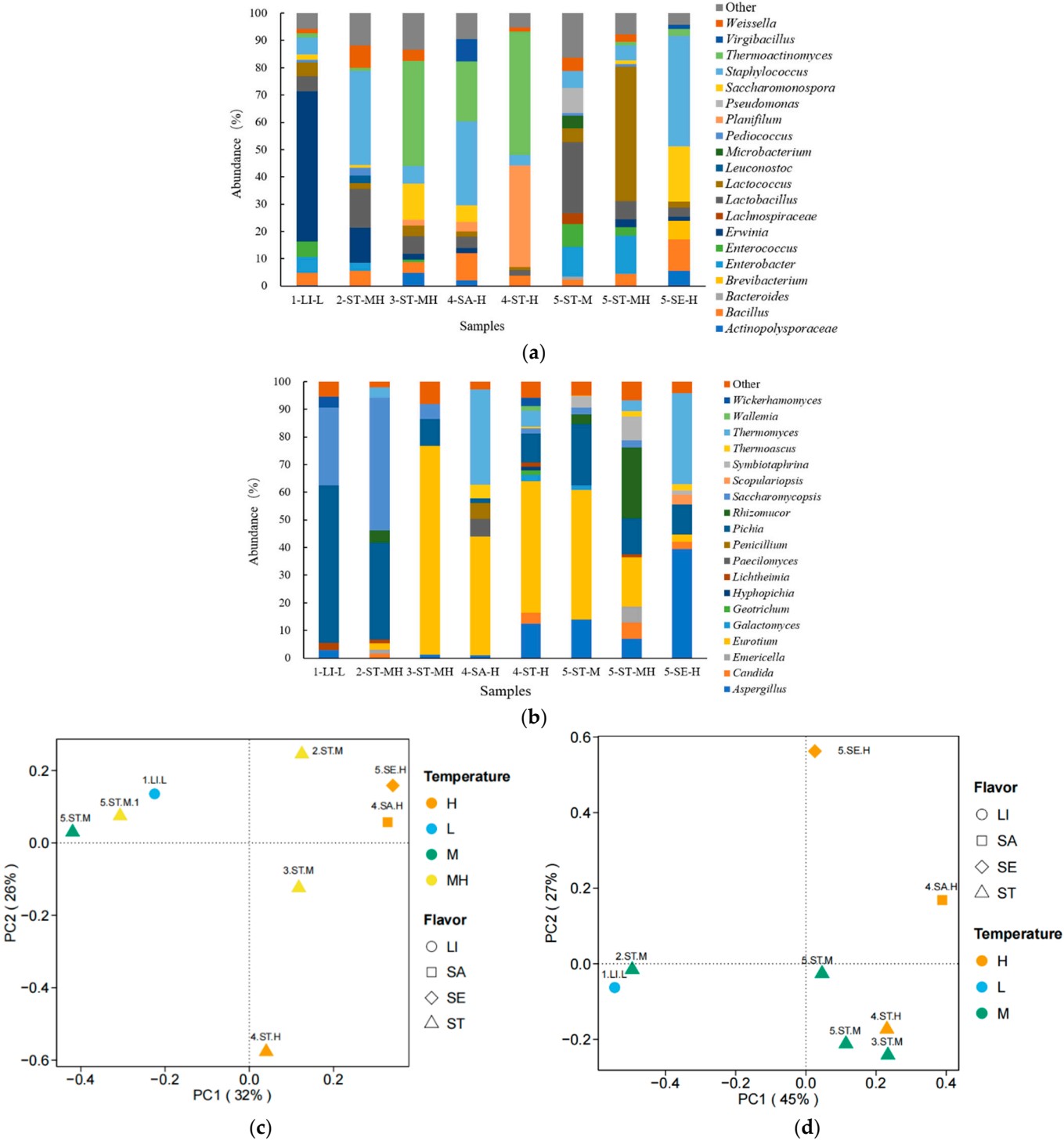

**Figure 2.** Composition and microbial diversity of 8 kinds of *Daqu* samples. (**a**) Bacterial community composition; (**b**) Fungal community composition; (**c**) PCoA of bacterial community; (**d**) PCoA of fungal community.

Taxonomic annotation of the OTUs generated by fungal ITS rRNA gene clustering in eight *Daqu* samples is shown in Figure 2b. A total of 71 OTUs belonged to 13 families and 30 genera (all OTUs were identified at the genus level, and 65 OTUs were identified at the species level). Among them, 19 OTUs with an abundance exceeding 1% were identified. The dominant fungal genus in *Daqu* samples was simpler than the bacterial community, including *Pichia*, *Saccharomycopsis*, *Eurotium*, *Thermomyces*, *Rhizomucor*, and *Aspergillus*. Of

them, *Pichia* and *Saccharomycopsis* were the predominant fungi in 1-LI-L and 2-ST-MH. However, *Thermomyces* accounted for the predominant genus in the 3-ST-MH, 4-SA-H, 4-ST-H, and 5-ST-M samples. The high abundance of high-temperature-resistant microorganism *Thermomyces* in those samples could be related to the relatively high temperature during *Daqu* production. The results of the PCoA analysis of the fungal community structure are shown in Figure 2d. The fungal community structure of the 4-SA-H and 5-SE-H sample differed the most from 1-LI-L (light-flavor *Daqu*). The PCoA results of both the bacterial and fungal communities demonstrated that 4-SA-H and 5-SE-H was the optimal *Daqu* sample for stacking fermentation.

Diversity indexes including Shannon index, Simpson index, ACE and Chao index were also taken into consideration during *Daqu* selection. Both the ACE index and the Chao index were used to estimate the number of OTUs in the microbial community and characterize the abundance of microorganisms in the sample; The Shannon index and the Simpson index were used to characterize the diversity of microorganisms in the sample. Results of the OTU clustering of the bacterial 16S rRNA genes in the samples were statistically analyzed, and the diversity results are shown in Table 2. A total of 101 OTUs were clustered in eight samples, of which 5-ST-M contained the largest number of OTUs. 5-ST-M had the highest number of OTUs, the highest Shannon index, and the lowest Simpson index. This indicates that it had the highest bacterial richness and the most diverse bacterial community. The Shannon index indicated that 2-ST-MH, 3-ST-MH, 4-SA-H, and 5-ST-M all have a more diverse bacterial community and were more suitable *Daqu* for stacking fermentation. Similar to the bacterial amplicon results, the sequencing coverage of all eight samples exceeded 99.96%, indicating sufficient sequencing depth. For the fungal community in Table 3, eight samples clustered 71 OTUs, and 4-SA-H contained the most OTUs. From the Shannon index, the sample with the highest fungal diversity index was 5-ST-MH, while the lowest value was 3-ST-MH. Based on ACE and Chao index of bacterial and fungal community, 4-SA-H processed the higher community richness than 5-SE-H. From the perspective of fungal and bacterial community structure, diversity, and richness, 4-SA-H was the ideal *Daqu* sample for stacking fermentation.

**Table 2.** The bacterial 16S rRNA gene diversity in different *Daqu* samples.

| *Daqu* Samples | Coverage/% | OTU | ACE | Chao | Shannon | Simpson |
|---|---|---|---|---|---|---|
| 1-LI-L | 99.98 | 67 | 81.42 | 79.50 | 2.03 | 0.33 |
| 2-ST-MH | 99.98 | 65 | 79.74 | 82.50 | 2.47 | 0.18 |
| 3-ST-MH | 100.00 | 66 | 74.40 | 74.00 | 2.64 | 0.14 |
| 4-SA-H | 99.99 | 62 | 71.53 | 70.20 | 2.59 | 0.14 |
| 4-ST-H | 99.98 | 56 | 74.94 | 74.14 | 1.55 | 0.34 |
| 5-ST-M | 100.00 | 94 | 95.26 | 95.00 | 3.19 | 0.07 |
| 5-ST-MH | 99.99 | 65 | 72.79 | 75.00 | 2.13 | 0.28 |
| 5-SE-H | 99.98 | 58 | 76.48 | 85.00 | 2.21 | 0.22 |

**Table 3.** The fungal ITS rRNA gene diversity in different *Daqu* samples.

| *Daqu* Samples | Coverage/% | OTU | ACE | Chao | Shannon | Simpson |
|---|---|---|---|---|---|---|
| 1-LI-L | 99.98 | 48 | 54.19 | 44.75 | 1.42 | 0.39 |
| 2-ST-MH | 99.96 | 52 | 60.70 | 59.00 | 1.39 | 0.36 |
| 3-ST-MH | 99.96 | 45 | 52.66 | 47.86 | 0.79 | 0.66 |
| 4-SA-H | 99.96 | 64 | 67.01 | 70.20 | 1.49 | 0.32 |
| 4-ST-H | 99.99 | 62 | 63.14 | 61.50 | 2.23 | 0.26 |
| 5-ST-M | 99.96 | 53 | 60.62 | 74.50 | 1.56 | 0.31 |
| 5-ST-MH | 99.98 | 60 | 55.91 | 53.75 | 2.41 | 0.14 |
| 5-SE-H | 99.98 | 49 | 51.83 | 51.00 | 1.77 | 0.26 |

### 3.2. Microbial Dynamics during Stacking and Alcoholic Fermentation

In this study, different selective media were used to reveal the microbial dynamics of total aerobic bacteria (TMAB), lactic acid bacteria (LAB), yeasts, and filamentous fungi of samples during the fermentation process in Figure 3a. During the first 24 h of stacking, the number of yeast and filamentous fungi increased sharply from 4.8 log CFU/g and 3.5 log CFU/g and decreased slightly in later stages of stacking. Changes in the number of yeast and filamentous fungi were different from those of bacteria. During stacking process, the amount of TMAB and LAB increased continuously, ranging from 0.5 log CFU/g to 1.3 log CFU/g. After adding original *Daqu*, the number of TMAB and LAB significantly increased and stabilized at approximately 8.0 log CFU/g and 7.5 log CFU/g, respectively. During the alcoholic fermentation stage, the growth rate of yeast increased, and peaked at 7.0 log CFU/g on the 12th day of fermentation, after which it remained relatively stable. Filamentous fungi also experienced a period of vigorous reproduction and peaked at 5.0 log CFU/g on the sixth day, though it decreased as fermentation progressed and continued to decrease until fermentation was below 3.0 log CFU/g. In summary, the number of bacteria is significantly higher than the number of fungi and the number of yeasts in fungi is significantly higher than that of filamentous fungi throughout the fermentation process.

To identify the diversity and taxonomy of the microbial community during the fermentation process, we performed amplicon sequencing on the 16S and ITS rRNA gene and found that a total of 174 bacterial OTUs belonged to 41 bacterial genera. Genera with a relative abundance exceeding 1% were considered dominant (Figure 3b). During early stacking fermentation stages, *Bacillus* (38.55%) and *Enterococcus* (31.54%) were the dominant bacterial genera. As the stacking fermentation progressed, the abundance of these two genera gradually decreased and in the alcoholic fermentation stage, it continued to decrease to less than 5%. *Lactobacillus* is the most important dominant microorganism in alcoholic fermentation, and its abundance reached 90% in F4, after which it decreased and stabilized at around 80%. *Pediococcus* became the main dominant microorganism after accumulation, with abundance reaching 50%. As fermentation progressed, it decreased to 2.25% in the F4 sample and stayed less than 10% in later fermentation periods. The PCoA analysis of the bacterial community structure during the fermentation process is shown in Figure 3d. Sample S1 was far from sample S3, suggesting the differential community structure. It indicates the main components of bacteria in the fermented grains have changed during stacking fermentation. The S3, F1, F2, and F3 samples were far from other samples, which demonstrates the effects of stacking on the bacterial community structure of fermented grains. F4, F5, F6, and F7 were clustered, indicating that the bacterial community structure remained stable from day 10 to 26 during the alcoholic fermentation stage.

For fungi, amplicon sequencing revealed a total of 133 OTUs belonging to 57 fungal genera. Genera with a relative abundance exceeding 1% were considered dominant (Figure 3c). The fungal diversity of fermented grains was lower than the bacterial diversity during fermentation. *Pichia* was the dominant genus during the fermentation process, with a relative abundance of approximately 50–90%, followed by the fungus *Saccharomyces*, with a relative abundance of approximately 5–21%. While the relative abundance of *Pichia* and *Saccharomyces* shifted during fermentation, the total abundance of *Pichia* and *Saccharomyces* remained stable, which suggested competition between the two dominant fungal genera. The PCoA analysis of the fungal community structure during the fermentation process is shown in Figure 3e. Sample S1 and S2 were separated from other samples, suggesting fungal community was changed during stacking process. F5, F6, and F7 were clustered, indicating that the fungal community structure remained stable from day 15 to 26 during the alcoholic fermentation stage.

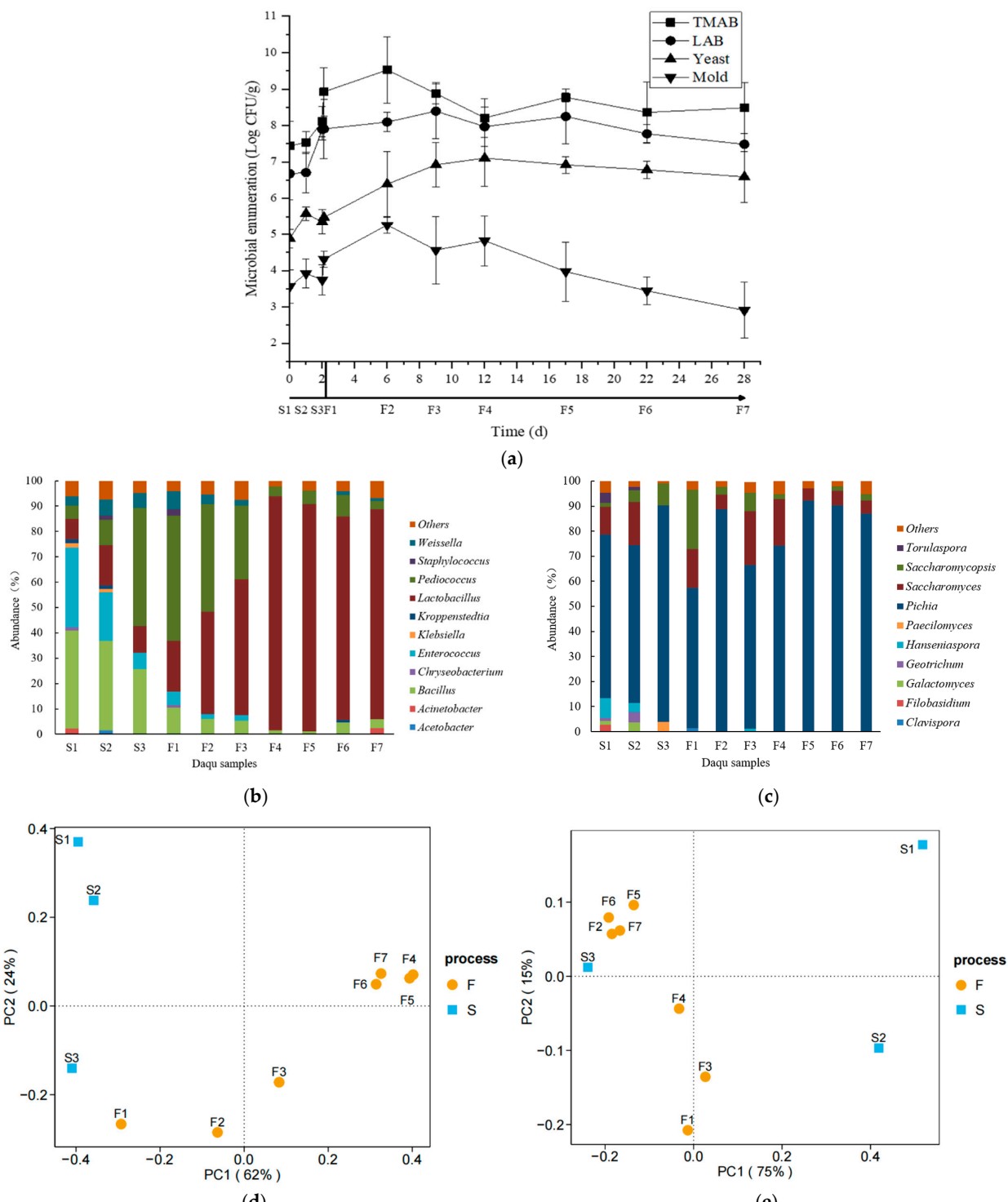

**Figure 3.** Microbial dynamics during alcoholic fermentation with stacking. (**a**) Microbial dynamics based on culture-dependent methods; (**b**) Bacterial community composition; (**c**) Fungal community composition; (**d**) PCoA analysis of bacterial community; (**e**) PCoA analysis of fungal community.

Differences in the composition of the bacterial and fungal microbial community between stacking *Daqu* and S3 highlight the effects of environmental screening or selection on diverse species, including increases in low-abundance genera such as *Lactobacillus* and *Pichia* and decreases in dominant genera such as *Staphylococcus* and *Thermoactinomyces*. Most dominant genera in S3 were highly abundant during alcoholic fermentation. The fungal community structure of S3 was more similar to the original *Daqu* than the bacterial

community, suggesting that environmental changes have a greater impact on fungi during the stacking process.

### 3.3. Physiochemical Properties during Stacking and Alcoholic Fermentation

To better understand the alcoholic fermentation process and analyze the correlation between microbial dynamics and environmental changes, physiochemical properties such as temperature, moisture, and pH were detected. A thermometer was used to continuously record the temperature of each part of the fermented mash. After the fermentation was completed, the data was imported into a computer for analysis; the results are shown in Figure 4a (left two coordinate axes). In the first 48 h of stacking, the temperature rapidly increased from 15 to over 45 °C, which is much higher than the temperature (less than 30 °C) of traditional light-flavor *Baijiu* fermentation and is similar to the production of low-temperature *Daqu*. After the second cold dispersion and the addition of low-temperature *Daqu*, the temperature of the fermented grains is similar to the beginning of alcoholic fermentation in traditional light-flavor *Baijiu*, and peaked at 27 °C on the seventh day of alcoholic fermentation (the ninth day of fermentation). It then slowly decreased to approximately 20 °C by the end of the fermentation process.

Near-infrared spectroscopy was used to detect the water content, and the results are shown in Figure 4a (left coordinate axis). During the first 48 h of stacking, the water content of the fermented grains was maintained at 47%, which fluctuates slightly. When alcohol fermentation began, the water content sharply increased to more than 50% on the sixth day of fermentation and remained at 50–57% throughout the alcoholic fermentation process.

Near-infrared spectroscopy was used to determine the starch, sugar, acidity, and alcohol content during fermentation, and the results are shown in Figure 4a (right coordinate). The starch content in the mash steadily decreased from 28 to 25% for 48 h during stacking, and the sugar content gradually increased. This indicates that the microorganisms produced a large amount of amylase during the stacking process. During alcoholic fermentation, the starch content sharply decreased to 20% on the sixth day, and then stabilized at 18% until the end of alcoholic fermentation. However, the sugar content significantly decreased on the sixth day and then stabilized at 5 mg/100 g. During the accumulation stage, the alcohol content increased at S3 and continued to increase to 10% as alcohol fermentation progressed. F3 was highest on the seventh day of alcohol fermentation and stabilized until the end of fermentation. The acidity was approximately 4.5 mg/100 g at the beginning of the stacking fermentation process, and decreased to 3 mg/100 g by the end and reached a maximum of 13.5 mg/100 g by the end of the alcoholic fermentation.

Spearman correlation analysis revealed the correlation between physical and chemical indexes and the microbial community in the fermentation process (Figure 4b). Bacterial genera strongly correlated with physical and chemical indexes include *Pediococcus*, *Chryseobacterium*, *Bacillus*, *Enterococcus*, *Weissella*, *Staphylococcus*, *Lactobacillus*, and *Kroppenstedtia*, while the physical and chemical indexes are concentrated in four indexes: alcohol content, acidity, starch content, and sugar content. The alcohol content was positively correlated with *Lactobacillus*, indicating that *Lactobacillus* had relatively high ethanol tolerance. The lactic acid produced by its metabolism can improve the acidity of fermented grains. The negative correlation between *Lactobacillus* and starch content indicates that starch can be used by *Lactobacillus* after degradation into small molecular sugar. *Staphylococcus* was negatively correlated with acidity, while *Bacillus* was negatively correlated with alcoholicity, indicating the effects of environmental factors on bacteria. *Chryseobacterium*, *Bacillus*, and *Enterococcus* were negatively correlated with alcohol content, indicating that as the alcohol content increased in fermented grains, the bacteria of these three genera became sensitive to ethanol and could not grow and reproduce normally; their abundance significantly decreased as a result. The correlation between the fungal community and physical and chemical indexes was weaker than that of the bacterial community (Figure 4c); there are only three fungal genera with significant correlations: *Pichia*, *Filobasidium*, and *Geotrichum*. *Pichia* was positively correlated with acidity, water content, alcohol content, temperature,

and other indicators of fermented grains, indicating that the microbial metabolism of this genus is vigorous and produces a large number of metabolites.

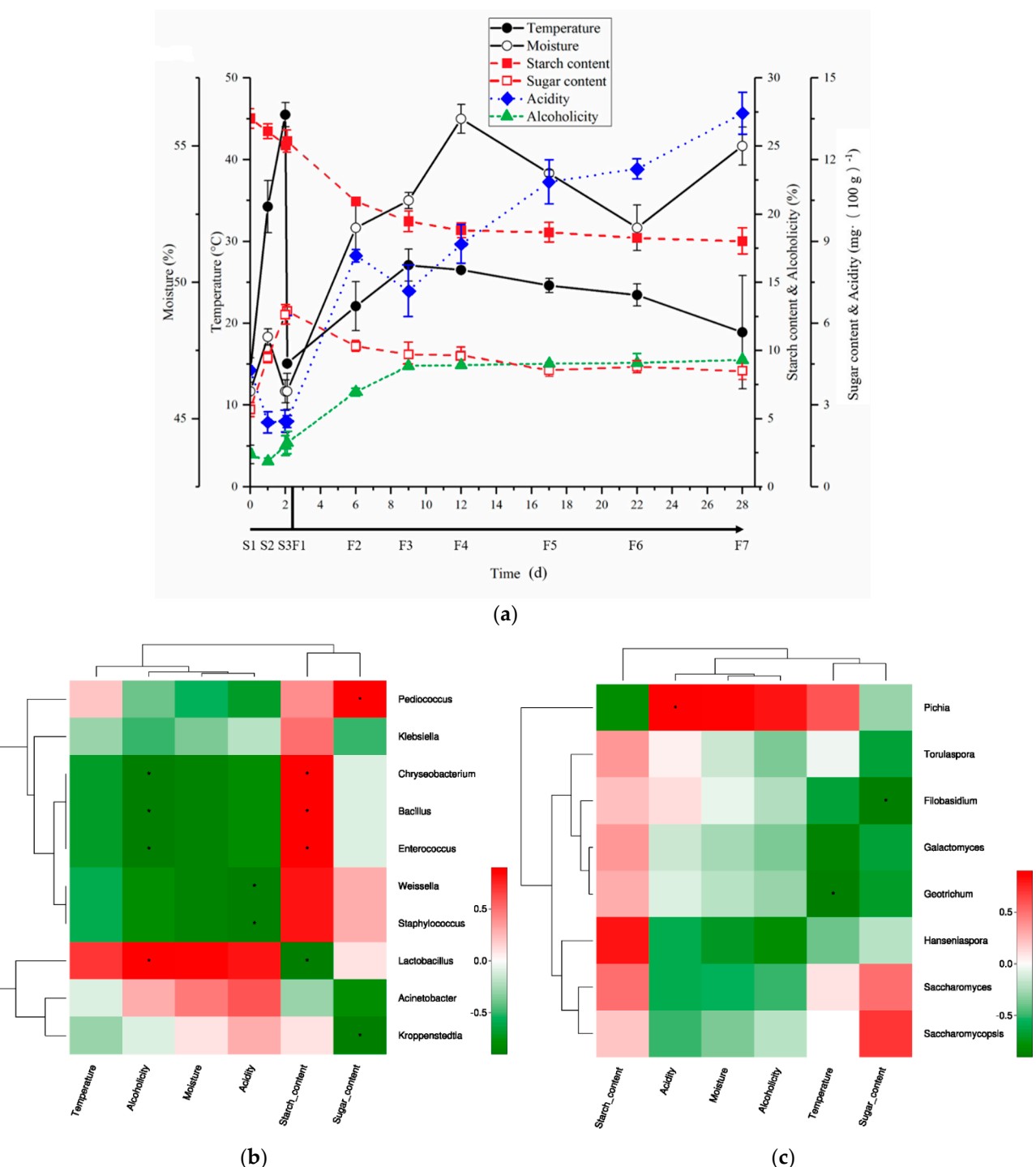

**Figure 4.** Physiochemical properties during alcoholic fermentation with stacking. (**a**) Physiochemical properties during alcoholic fermentation with stacking; (**b**) Correlation of physiochemical properties and bacterial community; (**c**) Correlation of physiochemical properties and fungal community.

Dominant microorganisms such as *Lactobacillus* and *Pichia* were positively correlated with temperature (15–45 °C), while the dominant genera in high-temperature *Daqu*, *Staphylococcus,* and *Bacillus*, were negatively correlated with temperature. This demonstrates that the temperature in the stacking process was suitable for *Lactobacillus* and unsuitable for

the growth of *Staphylococcus* and *Bacillus*. This explains the change in community structure from stacking *Daqu* to S3. The maximum temperature during stacking is approximately 45 °C, which is similar to the highest temperature for the production of the original low-temperature *Daqu*, though it is far below the maximum temperature of high-temperature *Daqu* stacking. Due to environmental selection, even without adding the original *Daqu* (1-LI-L), the fungal community structure of stacking was similar to 1-LI-L

### 3.4. Metabolite Dynamics during Alcoholic Fermentation with Stacking

Changes in flavor substances in fermented grain samples were detected at each stage of the fermentation process using HS-SPME-GC/MS technology. Percentage and concentration changes of volatile metabolites were presented in Figure 5a,b, respectively. As fermentation progresses, the relative abundance of acids and esters metabolites continued to increase and the proportion of alcohol metabolites was relatively stable (peaking in the late stage of stacking fermentation and decreasing after the start of the alcoholic fermentation stage), while the relative abundance of the other types of metabolism continues to decrease. During stacking fermentation, alcohols was the dominant volatile metabolites with the concentration over 60.0 ppm, while the concentration of acids and esters was below 20.0 ppm. As the alcoholic fermentation proceeds, the relative percentage of acids and esters significantly increased from 17.42 to 29.41%. The relative percentage of alcohols decreased sharply on the fourth and tenth day of alcoholic fermentation and stabilized around 50% after the 15th day of alcoholic fermentation. The concentration of alcohols decreased from 96.21 to 64.53 ppm during the first seven days of alcoholic fermentation, then increased to 125.20 ppm in the tenth day of fermentation. It finalized as 169.43 ppm at the end of the fermentation. The total ester concentration of liquor with stacking fermentation is significantly higher than that of ordinary light-flavor *Baijiu* (Figure 5d), demonstrating that the stacking process can increase the concentration of flavor substances during the traditional fermentation of light-flavor *Baijiu*, promote the accumulation of esters, and improve liquor quality. As shown in Figure 5c, the principal component analysis of the volatile flavor substances in the fermented grain samples at each fermentation stage demonstrates that the difference between S1 and S3 is caused by stacking fermentation, and that the material difference between F1 and S1 reflects the influence of the stacking process on alcoholic fermentation on flavor substances. F3, F4, F5, F6, and F7 are clustered together, which indicates that volatile flavor substances tend to be stable starting from F3 (the seventh day of alcoholic fermentation), and that flavor substances in conventional light-flavor *Baijiu* fermentation without stacking fermentation tend to be stable on the 15th day. This demonstrates that stacking fermentation can significantly shorten the time needed for alcoholic fermentation [22].

After analyzing the correlation between microbes and volatile flavor data through O2PLS, data with absolute correlation coefficient values greater than 0.7 were visualized (Figure 6). Microorganisms with an absolute VIP value of more than 0.7 come from five bacterial genera and three fungal genera, including *Lactobacillus*, *Pediococcus*, *Bacillus*, *Enterococcus*, *Pichia*, *Saccharomycopsis*, *Weissella*, and *Hanseniaspora*. Except for *Weissella* and *Staphylococcus*, almost all microorganisms are positively correlated with flavor substances to varying degrees. Ethyl acetate (A15) is an important flavor substance in traditional light-flavor *Baijiu* and is positively correlated with *Lactobacillus* and negatively correlated with *Weissella* and *Staphylococcus*.

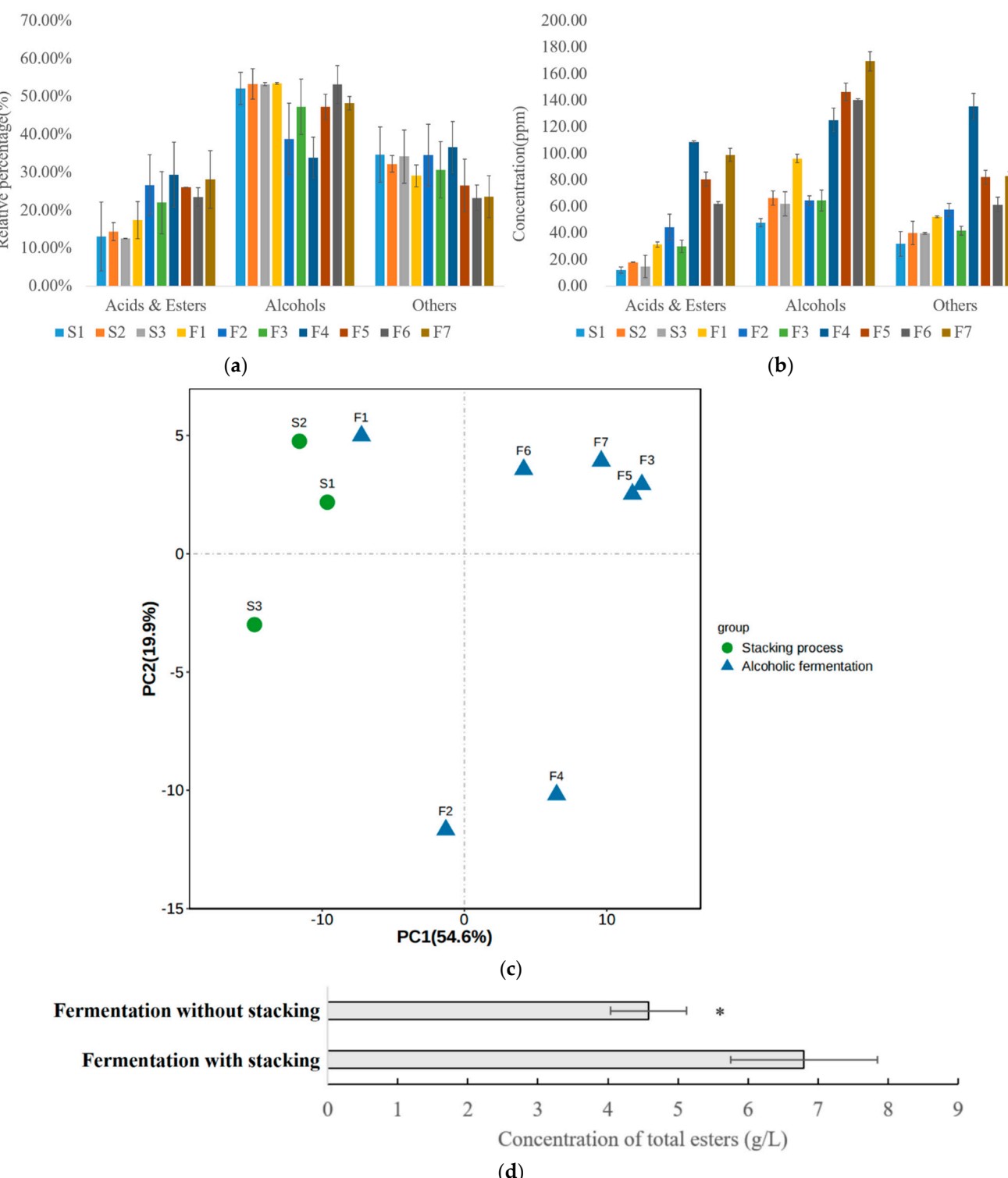

**Figure 5.** Volatile metabolite dynamics during alcoholic fermentation with stacking process. (**a**) Relative percentage of volatile metabolites; (**b**) Concentration of volatile metabolites; (**c**) PCoA analysis of volatile metabolites; (**d**) The total ester concentration of final liquor with stacking process (* means a significant difference between these two samples, $p < 0.05$).

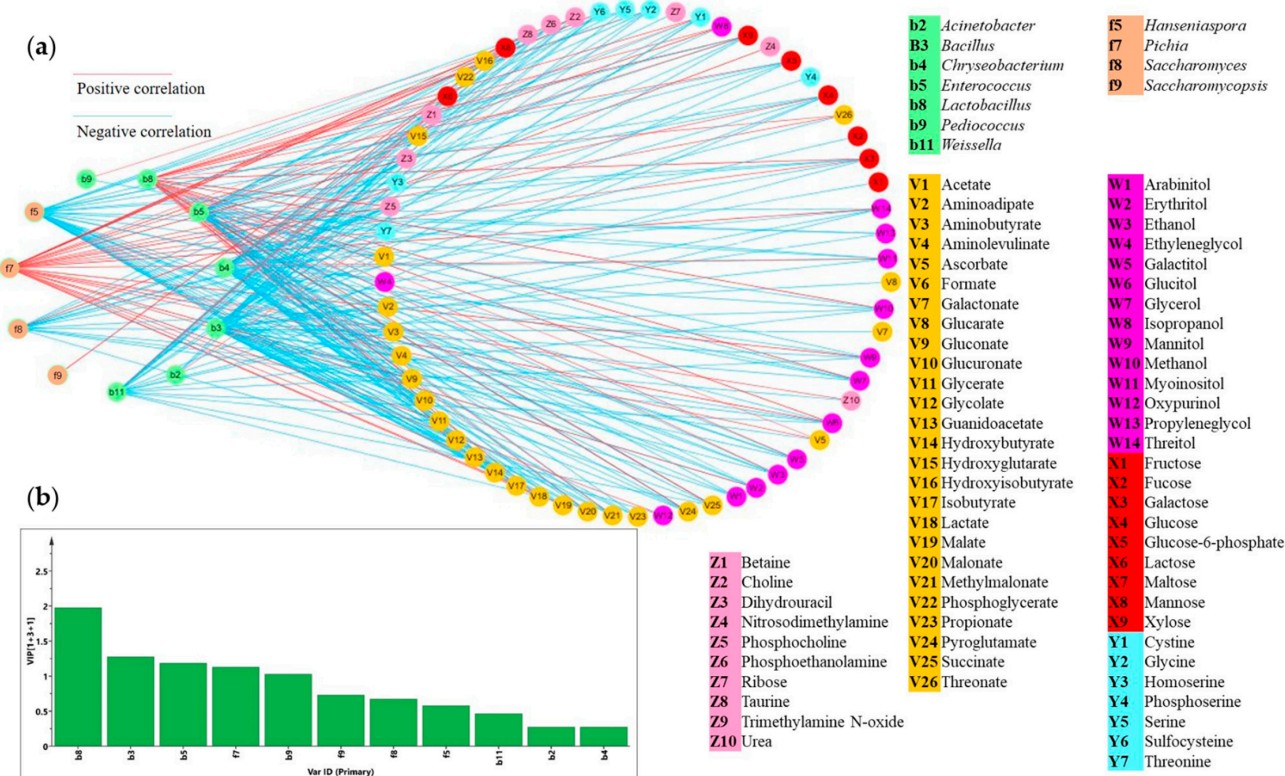

**Figure 6.** Correlation of microbial community and metabolites by O2PLS during fermentation. (**a**) The correlated network between genera and metabolites; (**b**) VIP plot of different genera ($|\rho| \geq 0.7$).

NMR detected 66 kinds of metabolites, including 26 kinds of acids and esters, 14 alcohols, 9 sugars, 7 amino acids, and 10 other compounds (primarily nitrogen-containing compounds) as in Figure 7. The content of sugars represented by fructose, trehalose, lactose, maltose, and mannose remained stable throughout the fermentation process, indicating that the consumption and utilization rate of these sugars by microorganisms is the same as the enzymatic hydrolysis rate and that the concentration is maintained in a dynamic balance. The concentration of galactose, glucose, and xylose continued to increase as fermentation progressed and showed a tendency to accumulate. Acids and esters have always been important flavor substances in fermented products, especially liquor. The primary acids and esters produced by metabolism accumulated, which not only increases the acidity of the fermented grain and inhibits the growth of certain microorganisms, but also forms the unique flavor of light-flavor *Baijiu* [5]. This demonstrates the functions of acetic acid bacteria and lactic acid bacteria during the fermentation process from the perspective of metabolites. The most important target product of liquor fermentation is ethanol. The ethanol concentration significantly increases during the early stages of fermentation and remains relatively stable in the middle stage, but declines in later stages since ethanol reacts with acid to form corresponding esters.

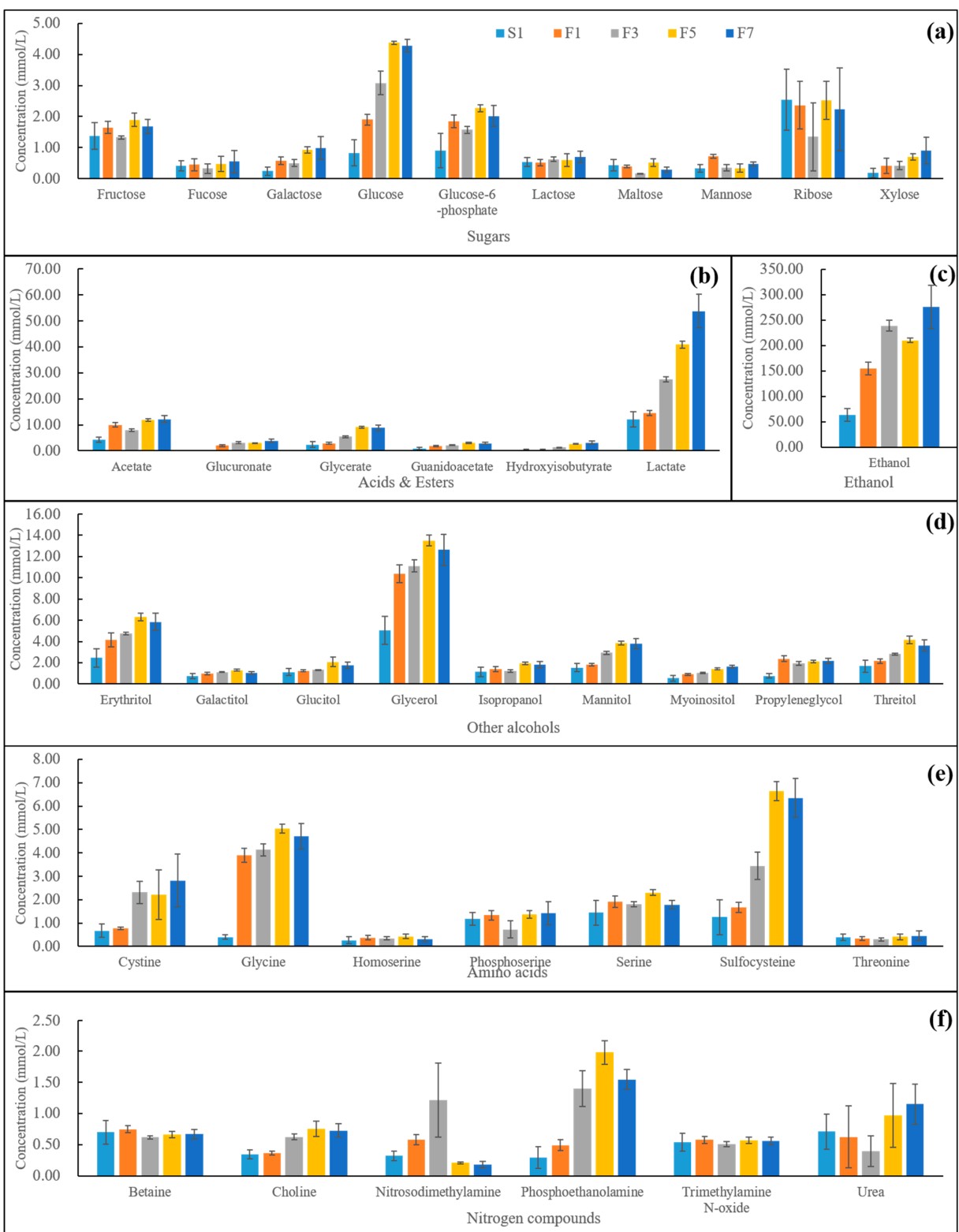

**Figure 7.** Dynamic of non-volatile metabolites during fermentation. (**a**) sugars concentration; (**b**) acids and esters concentration; (**c**) ethanol concentration; (**d**) other alcohols concentration; (**e**) amino acids concentration; (**f**) nitrogen compounds concentration.

From S1 to F1, the concentration of metabolites such as sugar, glucose, acetic acid, and ethanol significantly increased, which highlights the effects of stacking fermentation. The accumulation of ethanol and acetic acid can inhibit microbial growth during the alcoholic

fermentation process and promote the accumulation of flavor substances. A Spearman correlation analysis was used to analyze the correlation coefficients between microbial communities and metabolites during the fermentation process, and the correlation coefficient matrix was used to draw a heatmap (Figure S2). *Lactobacillus* is positively correlated with metabolites, while *Klebsiella*, *Chryseobacterium*, *Bacillus*, *Enterococcus*, *Weissella*, and *Staphylococcus* are negatively correlated with metabolites. The metabolites involved are primarily concentrated in substances closely related to cell metabolisms, such as lactic acid, gluconic acid, taurine, glucose, glycine, erythritol, glycerol, 6-phosphate glucose, and glyceric acid. This demonstrates that *Lactobacillus* plays an important role in the accumulation and consumption of metabolites, indicating that this dominant bacterial genus is relatively large and plays an important role in the fermentation process. Fungal genera that are positively and negatively related to metabolites are *Pichia* and *Hansenlaspora*, respectively, and there is little difference in the components of metabolites that are closely related to fungi and bacteria. This indicates that *Pichia* plays a similar role to *Lactobacillus* in regulating metabolite concentration during the fermentation process. Changes in volatile flavor substances and metabolites during the fermentation process indicate that the stacking process can provide microbial strains for alcoholic fermentation to regulate microbial community structure as well as enzymes, sugars, and flavor precursors, contribute to saccharification, and promote the production of flavor compounds.

## 4. Discussion

This study explored applications for stacking fermentation during the production of light-flavor *Baijiu* production and found that stacking fermentation can significantly increase the concentration of flavor substances. Stacking with high-temperature *Daqu* introduced a diverse range of microorganisms into the alcoholic fermentation system, which significantly altered the composition of the bacterial community during the first two days of stacking fermentation, particularly bacteria related to the production of flavor compounds such as *Bacillus* and *Lactobacillus*. The microbial community remained stable after 7–20 days of alcoholic fermentation, while the concentration of total esters was significantly higher than liquor produced without stacking process during fermentation. Flavor compounds such as esters and acids were positively correlated with dominant genera such as *Lactobacillus*, *Bacillus*, and *Pichia*, indicating that stacking promotes the production of flavor compounds by regulating the structure of the bacterial community and providing abundant carbohydrates (such as glucose) for fermentation. This demonstrates the versatility of stacking fermentation and reveals the mechanism of stacking during alcoholic fermentation, which entails the environmental selection of functional microorganisms and the accumulation of nutrients, promotes the fermentation process, and increases the concentration of certain flavor compounds. As such, the stacking process can be used to strengthen the flavor of different kinds of *Baijiu*.

The stacking process helps microorganisms adapt to a sorghum-based fermentation environment while stacking *Daqu* provides a microorganism source and the environment determines the microorganisms that can easily survive. Therefore, the S3 bacterial community exists in two kinds of *Daqu*. The S3 fungal community is similar to that of the original *Daqu* (1-LI-L), which is a relatively low-temperature environment. Low-abundance genera in high-temperature *Daqu* (4-SA-H) such as *Lactobacillus* and *Pichia*, became the dominant microorganism in the stacking environment. This is similar to original *Daqu*, in which similar microorganisms have been screened out in similar environments. Previous studies have demonstrated that environmental factors primarily affect the microbial community structure of *Daqu*. Environmental factors are the main reason for the transition of microbial community structures during thermophilic fermentation stages, particularly moisture, pH, acidity, and temperature [15]. *Daqu* production is affected by environmental factors such as acidity, moisture content, and temperature [23], which can explain the differential community structures among various *Daqu* [24,25]. In this study, the diverse microbial compositions in eight *Daqu* samples were related to temperature. The abundance

of thermophilic microorganisms, including *Thermoactinomyces* and *Thermomyces*, was higher in high-temperature *Daqu* samples, while the abundance of lactic acid bacteria and other microorganisms that are not resistant to high temperatures was higher in medium- and low-temperature *Daqu* samples. This is consistent with previous studies that used different methods [25–28]. During stacking and alcoholic fermentation, *Lactobacillus* abundance was positively correlated to temperature. The temperature during the stacking process was suitable for *Lactobacillus* growth, but not for the growth of dominant bacteria *Staphylococcus* and *Thermoactinomyces* in high-temperature *Daqu*. This indicates that though stacking *Daqu* is the microbial resource, the microbial community structure during stacking fermentation is mainly determined by environmental factors. During stacking, low abundance functional microorganisms in high-temperature *Daqu* were screened out, enriched, and amplified, while microorganisms not found in stacking *Daqu* were found in samples S1–S3. This demonstrates that some microorganisms in the fermentation process can also come from raw materials or the surrounding environment. Few previous studies have reported the contribution of microorganisms from raw materials to alcoholic fermentation [29–31]. The fungal species *Saccharomycopsis fibuligera* and *Pichia kudriavzevii* in *Daqu* primarily originated from associated tools and indoor surfaces [29–31]. According to our results, dominant fungi, including *Hanseniaspora* and *Saccharomycopsis*, largely came from raw materials or the environment.

In addition to the microbial community, the stacking process also affected physicochemical parameters and metabolites in fermented grains during alcoholic fermentation. We compared our results with results of normal light-flavor *Baijiu* fermentation obtained by previous studies [20,22,32]. Stacking fermentation increased temperatures during alcoholic fermentation under the similar temperature patterns. The normal temperature peak on the seventh day is approximately 20 °C and stabilizes around 15 °C by the end of the fermentation process [20,30,33]. The stacking process increased temperatures to 26 °C on the seventh day and approximately 18 °C by the end of alcoholic fermentation, which could be one explanation for flavor accumulation. Previous studies found that the composition of the microbial community and flavor compounds were stable on the seventh day of alcoholic fermentation [22,34,35]. After the stacking process, the fungal community remained stable from the fourth day of alcoholic fermentation, indicating that stacking increased the adaptation time for functional fungi during the first four days of fermentation. A recent study demonstrated that *Lactobacillus* and *Saccharomyces cerevisiae* formed a multi-species biofilm to survive in *Daqu* and required certain time and nutrients to transform from biofilm to a planktonic state [21]. Generally, *Lactobacillus* and *Saccharomyces cerevisiae* become the dominant species on the seventh day of alcoholic fermentation [20,30,33]. Also, *Lactobacillus* and *Saccharomyces* are the dominant microorganisms in spontaneous alcoholic fermentation of beer including American coolship ale [36] and Lambic beer [37]. In this study, the abundance of *Saccharomyces cerevisiae* sharply increased and became the dominant species after the fourth day of alcoholic fermentation, indicating that stacking promotes the conversion of biofilm to a planktonic state.

In conclusion, the stacking process is an effective method of improving the flavor and quality of *Baijiu*, while stacking *Daqu* only inoculates several microorganisms. The dominant genera during the stacking process were more similar to the ones during normal light-flavor *Baijiu* fermentation process. The liquor distilled from this study has a similar flavor to the light-flavor *Baijiu*. As such, the stacking process can be used to produce sauce-flavor liquor, and has wide application potential in *Baijiu* and other solid-state fermentation processes. Combined with the influence of environmental factors on community structure, light-flavor *Daqu* can be used for stacking fermentation to fully saccharify and improve the accumulation of flavor substances; however, additional study on the interactions between bacterial and fungal microbes during stacking fermentation is needed before this process can be widely adopted.

**Supplementary Materials:** The following supporting information can be downloaded at: https://www.mdpi.com/article/10.3390/fermentation8020067/s1, Figure S1: Rarefaction curves of samples. (a) bacterial 16S rRNA gene; (b) fungal ITS domain of rRNA gene; Figure S2. Correlation of microbial community and non-volatile metabolites during fermentation. (a) correlation of bacterial community and metabolites during fermentation; (b) correlation of fungal community and metabolites during fermentation.

**Author Contributions:** Conceptualization, Z.L.; Data curation, Z.L. and X.H.; Formal analysis, Z.L.; Funding acquisition, Z.L. and B.H.; Investigation, Z.L.; Methodology, Z.L.; Project administration, B.H.; Resources, Z.L.; Supervision, B.H.; Visualization, Y.F.; Writing—original draft, Z.L. and Y.F.; Writing—review and editing, Z.L., Y.F. and X.H. All authors have read and agreed to the published version of the manuscript.

**Funding:** This research was funded by National Natural Science Foundation of China (No. 31671829) and Specialized Research Fund for the Doctoral Program of Higher Education (No. 20130008110013).

**Institutional Review Board Statement:** Not applicable.

**Informed Consent Statement:** Not applicable.

**Conflicts of Interest:** The authors declare that the research was conducted in the absence of any commercial or financial relationships that could be construed as a potential conflict of interest.

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
