# Peer review of "Microbial Diversity and Metabolites Dynamic of Light-Flavor Baijiu with Stacking Process"

_fermentation, doi:10.3390/fermentation8020067_

Round 1

Reviewer 1 Report

Comments are attached.

Reviewer 2 Report

Manuscript fermentation-1560395

The manuscript investigates the microbial diversity and metabolites dynamic of light-flavour Baijiu with stacking process. For the study, advanced analytical techniques and statistical tools are used. Results are then an interesting contribution. However, interpretation of results is excessively simple and may lead to errors. A cautious interpretation of results is advised. Besides, the presentation should be improved (italic in names is chaotic), and the redaction could be made more fluid and repetitions prevented.

Some specific comments follow.  

Introduction. It is large and sometimes repetitive. It could be reduced and simplified.

Table 1. It could be simplified by removing flavour and province in the columns. Their mention in the headings could be enough. In the first columns, also acronyms could be introduced in brackets in the column heading.

L105 -118. The process should be described in more detail since most readers could not be familiar with it. For example, the proportion of water added, the meaning of ventilation, etc.

L 148. Cultures at 30º C for 3-4 d. Please, be concrete.

156 um  ?????

L 173-176. The meaning of the sentence is not clear.

L197. Please revise italic throughout the text.

L202. Include respectively, to indicate that each was abundant in a sample.

L204. 4-ST-H is also far from 1-LI-L

L229. The sample 5-SE-H is also far from 1-LI-L

L237. Some comment on ACE and Chao is expected.

L248. It is convenient to specify why the bacterial diversity is ideal

L255. It is unclear to what sample or mixture of samples, the data in figure 3a refer to

L266. Partial increase?

L284. Is it contradictory that the most abundant microorganism during the alcoholic fermentation was Lactobacillus?

L290-292. The explanations are hardly understood.

L306-308. Again, the explanations are hardly understood

Figure 4a is complex to understand. Maybe it could be split into two Figures.

L348-350. Do the data support such a hypothesis.

L351-363. The changes in percentages are of difficult interpretation since they are not independent. The increase in one component implies the decrease in another or various. Then, they should be assessed cautiously.

L378-401. The environment is essentially multivariate. Then, correlations could be not as simple as sometimes commented. E.g. correlation between Lactobacillus and Pichia with temperature could be reasonable, but why not the decrease in Staphylococcus and Bacillus was not due to the decrease in pH because of the increase in acidity (o decrease in pH) due to Lactobacillus activity? Be careful.

L404-435. The authors speak about components that increase or decrease in percentages. But the proportions are not independent. The increase in one component automatically implies a decrease in the other. Furthermore, the more rapid increase in one component than in another may create the false impression that the second is decreasing. Then, this reviewer is sceptic regarding this interpretation.

Round 2

Reviewer 2 Report

Authors have attended conveniently the suggestions and comments. Still a few comments. Table 1, the last column could be improved by indicating province in the heading, e.g. Location (province). L303 "clustering together" hardly reflects reality. They are far from others, but also somewhat separate among them; so "together" sounds excessive. In addition, still, there are some minor errors like the use of the capital letter, for example.

Author Response

Dear Reviewer:

    Thank you again for your rigorous comments, which are very helpful for revising and improving our paper. We have studied comments carefully and have made correction which we hope meet with approval. Revised portion are marked with different colors in the paper. The main corrections in the paper and the responds to the comments are as flowing:

1. Table 1, the last column could be improved by indicating province in the heading, e.g. Location (province).

Thanks so much for the suggestion. We indicated province in the heading in Table 1 in the manuscript.

2. L303 "clustering together" hardly reflects reality. They are far from others, but also somewhat separate among them; so "together" sounds excessive.

Thanks so much for the suggestion. We revised the sentences as "The S3, F1, F2, and F3 samples were far from other samples, which demonstrates the effects of stacking on the bacterial community structure of fermented grains." in L302-304.

3. In addition, still, there are some minor errors like the use of the capital letter, for example.

Thanks so much for the suggestion. The errors were corrected in L108, L159, L161, L166, L181, L183, L186, L313, L369, L408, L440, L556.